# Experimental and Simulation Studies on the Compressive Properties of Brazed Aluminum Honeycomb Plates and a Strength Prediction Method

**Lanxin Jiang [1]** , **Shoune Xiao [1]**, **Jingke Zhang [1]**, **Ruijuan Lv [1]**, **Bing Yang [1,\*]**, **Dawei Dong [2]**, **Guangwu Yang [1]** **and Tao Zhu [1]**

[1] State Key Laboratory of Traction Power, Southwest Jiaotong University, Chengdu 610031, China; jlx@my.swjtu.edu.cn (L.J.); snxiao@swjtu.edu.cn (S.X.); jkzhang@my.swjtu.edu.cn (J.Z.); lrj26@my.swjtu.edu.cn (R.L.); gwyang@swjtu.edu.cn (G.Y.); zhutao034@swjtu.edu.cn (T.Z.)
[2] Department of Thermal Power and Automotive Engineering, School of Mechanical Engineering, Southwest Jiaotong University, Chengdu 610031, China; dwdong@swjtu.edu.cn
\* Correspondence: yb@swjtu.edu.cn; Tel.: +86-28-86466433

**Abstract:** To study the compressive mechanical properties of a new type of brazed aluminum honeycomb plate (BAHP), tensile tests on single- and brazed-cell walls as well as compression tests in the out-of-plane, in-plane longitudinal, and transverse directions were conducted. Compared to the material properties of a traditional glued aluminum honeycomb plate (GAHP), those of the single- and brazed-cell walls of the BAHP are entirely different. Therefore, their characteristics should be considered separately when performing theoretical and simulation analysis. Under out-of-plane compression, the core of the BAHP did not debond, owing to its higher strength than that of the GAHP. In comparison, under in-plane compression in the longitudinal and transverse directions, the load–displacement characteristics, ultimate load, and failure modes also differed, and there was no large-scale cracking. Considering the characteristics of the BAHP, a strength prediction method was proposed. The simulation results demonstrated that the model built based on the new method was highly consistent with the experimental results. Defects with uneven height and debonding will cause the overall instability, and the degree of defects will influence the strength and instability displacement, which have little impact on the elastic stage. Moreover, the model considering defects is closer to the test results.

**Keywords:** brazed aluminum honeycomb plate; out-of-plane compression; in-plane compression

## 1. Introduction

Owing to their high strength and low weight, aluminum honeycomb sandwich structures have been extensively used in rail transportation, aerospace, and other fields, in recent decades, and have developed rapidly. The manufacturing process of aluminum honeycomb plates is also being continually developed. Typically, the core of the traditional aluminum honeycomb is glued to a panel to form a glued aluminum honeycomb plate (GAHP), which tends to easily debond during loading [1–3]. A new manufacturing method recently applied in engineering has led to the formation of brazed aluminum honeycomb plates (BAHPs). Using the brazing technique significantly increases the interfacial strength, however, the mechanical properties, failure mechanism, simulation method, and theoretical strength prediction need to be further studied.

At present, the research on aluminum honeycomb structures in academia is mainly concentrated on GAHPs, however, this can provide the basis and methods for studies on BAHPs. Bai et al. [4–6]

focused on a unit of the honeycomb core and by combining the folding element theory and the principle of energy conservation, considered the influence of an interfacial adhesive to theoretically predict the effects under out-of-plane compressions. Considering a Y-shaped structure, it was found that debonding has a significant influence on the mechanical properties of the core, and the debonding occurring on the outer edge is the most critical for a particular damage area. The above studies also found that when the adhesive strength is insufficient or the defect area is large, a peeling fracture can possibly occur. Problems during manufacturing, such as uneven distribution of the honeycomb, irregular hexagons, and debonding, are expected to significantly reduce the bearing capacity and energy absorption of a structure [7,8]. Studies on honeycomb cores provide a reference to investigate aluminum honeycomb sandwich plates. Rajkumar et al. [2,9–13] focused on their mechanical properties under tension, compression, and shear; analyzed and summarized the characteristics of load–displacement curves and failure modes; and proposed an equivalent model.

Simultaneously, Paik et al. [14] comprehensively summarized the properties under out-of-plane and in-plane compression and bending conditions and developed the corresponding strength prediction formulas and equivalent methods. In addition to compression properties, the bending properties of aluminum honeycomb plates [15–17] are also research topics attractive to numerous scholars. However, the mechanical properties, failure modes, and strength of a GAHP are quite different from those of a BAHP. As a new type of aluminum honeycomb plate, only limited research data are available on BAHPs. For example, Peng [15] conducted finite element analysis (FEA) to study the three-point bending properties of a BAHP as well as compared the effects of different unit cell thicknesses on the failure modes. Cai et al. [10] simply compared the peeling stresses of a GAHP and a BAHP and suggested that the strength of a brazed connection is better than glued one.

Therefore, it is necessary to study the mechanical properties and simulation methods of BAHPs. In this study, out-of-plane and in-plane compression tests are conducted on a BAHP to obtain the load–displacement curves and failure modes, which are different from those of a traditional honeycomb plate. Based on the deviations, a new theoretical method is proposed to predict the out-of-plane compressive strength. The finite element software (version 7.0, LSTC company, Livermore, CA, USA), LS-DYNA, is used to conduct the simulations of the BAHP based on the new and traditional methods.

## 2. Brief Introduction of BAHP and Honeycomb Core Experiment

### 2.1. Brazed Aluminum Honeycomb Plates (BAHPs)

A BAHP is a metal structural material plate composed of a core formed of a welded aluminum honeycomb, with aluminum plates coated on both its sides. Following assembly, it is placed in a brazing furnace and brazed at once, at 600 °C. As displayed in Figure 1, the heating temperature during welding is higher than the melting temperature of the powder solder and lower than the melting temperature of the base material. After the dissolution, diffusion, cooling, and solidification, the single-cell walls, core, and plates are tightly combined. A BAHP has the characteristics of high connection strength, because its drum peel and compressive strengths are better than those of a GAHP. Concurrently, all BAHPs are formed of pure aluminum alloys, without other non-metallic materials and any toxic or harmful substance during the process; therefore, they can be 100% recycled. Furthermore, the entire production chain is environmentally friendly, from the preparation of the raw materials to the recycling. The impact resistance and sound insulation performance of a BAHP also need to be further examined. In the honeycomb core, there are single-cell walls and double-cell walls, and the direction along the double-cell walls is longitudinal direction(L-direction), perpendicularly to the double-cell walls is transverse direction (T-direction), as shown in Figure 1.

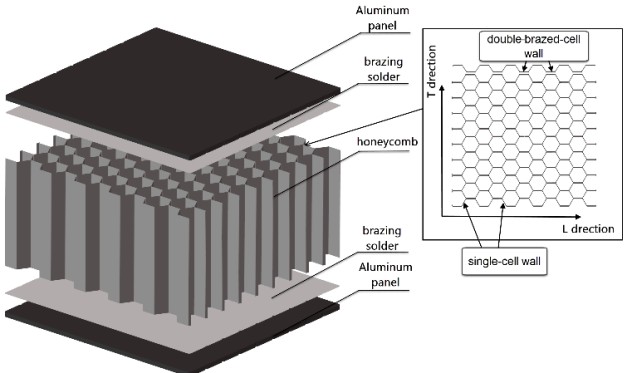

**Figure 1.** Model of a brazed aluminum honeycomb plate (BAHP).

*2.2. Test on Cell Wall in Core*

Based on the processing technology, the interfacial bonding of a BAHP core is different from that of a GAHP. To study the changes in the core material after brazing, tensile tests were conducted on one single-cell wall samples and ten-brazed-cell wall samples in a honeycomb. The specially designed ten-brazed-cell wall specimens can reflect the constitutive characteristics of aluminum alloy after welding. The test reference standards were GB/T 228.1-2010, "Tensile test for metal materials Part 1: Room temperature test method." Three samples of the two types of walls were tested. They had thicknesses of 3.2 mm in the ten-cell wall and 0.3 mm in the single-cell wall, with dimensions (length × width) of 143 mm × 20 mm. The test was performed on an MTS858-BIONIX tensile torsion tester (MTS company, USA) at a loading speed of 0.5 mm/min. The experimental data were collected by a Teststar II 490 Series computer system (Teststar company, Jinan, China), and the strain was measured by an extensometer with a gauge distance of 10 mm.

As can be seen from Figure 2a,b, the brazing process has a significant effect on the cell wall properties. The stress–strain curve of the single-cell wall is nearly bilinear, and its elastic and plastic phases are notably distinguishable. Following the brazing, the stress–strain curve becomes smooth, with no remarkable elastic–plastic boundary.

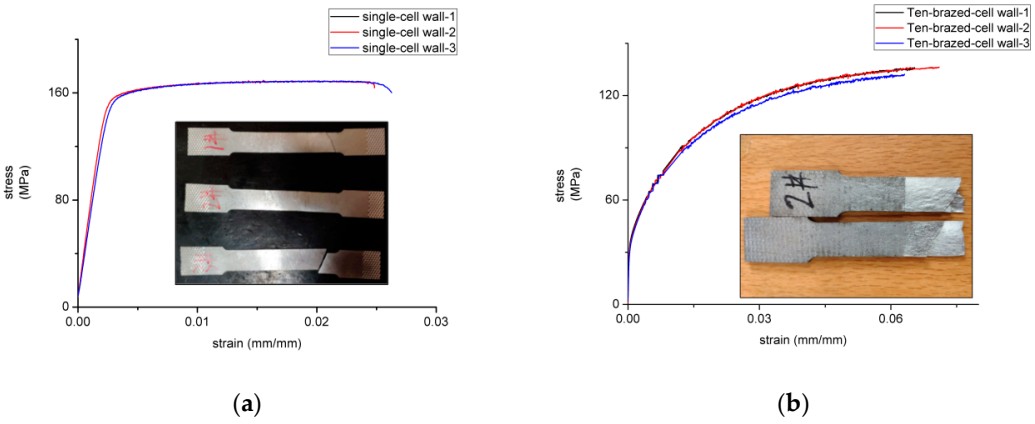

**Figure 2.** (**a**) Tensile test results of single-cell wall specimens; (**b**) Tensile test results of a ten-brazed-cell wall specimens.

The failure mode is an interlayer damage, and the ultimate stress is slightly smaller than that on a single cell. In addition to the thickness discrepancy, the material properties are also significantly different, based on the brazing comparison between the experimental and theoretical results. The modulus of the aluminum sheets increase and their yield points decrease after brazing. Therefore, the changes following brazing should be incorporated in simulations and theoretical analyses, and single- and double-cell walls are considered to be different in a honeycomb.

## 3. Tests on BAHP

### 3.1. Out-of-Plane Compression Test

When preparing a sample, the incision should be smooth and without burrs. The test referred to standard GBT1453-2005, "Test method for flatwise compression properties of sandwich constructions or cores". The lengths, widths, and thicknesses of the samples were 94, 94, and 50 mm each, respectively, and the thicknesses of the single-cell wall and double-brazed-cell wall were 0.2 mm and 0.6 mm, respectively. Specifically, the thickness of the double-brazed-cell wall was more than twice that of the single one, owing to the brazing solder. The brand of panels is A602, and the core is 3003. The compositions of the aluminum alloys are shown in Table 1. The test was performed on an MTS tester at a loading rate of 0.5 mm/min. The compression stress–displacement curves and structural failure modes are presented in Figure 3. The main failures are buckling and squeezing of the core in the middle, without cracks in the brazing area of the panel and double-cell wall.

**Table 1.** The material compositions of panels and core.

| Construction | Brand | Si | Cu | Mn | Zn | Fe | Al |
|---|---|---|---|---|---|---|---|
| Panel | A602 | 0.5–1.2% | 0.2–0.6% | 0.15–0.35% | 0.2% | 0.5% | the rest |
| Core | 3003 | 0.6% | 0.05–0.2% | 1.0–1.5% | 0.1% | 0.7% | the rest |

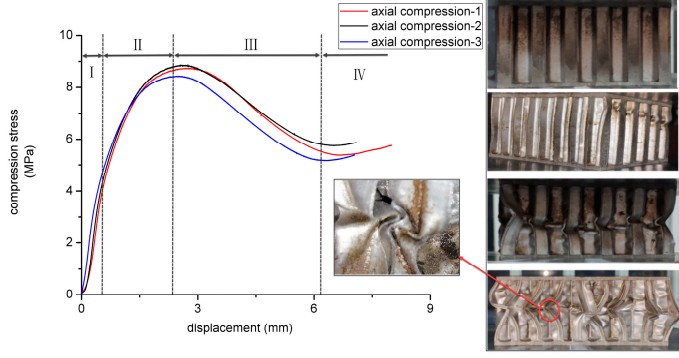

**Figure 3.** Stress–displacement curves and failure modes of the BAHP.

As shown in Figure 3, the out-of-plane compression of the BAHP comprises an elastic buckling stage, a plastic buckling stage, a honeycomb instability stage, and a dense strengthening stage. During the first stage, elastic buckling is due to the lateral pressure, which then develops into plastic buckling with increasing load. When the maximum load is reached, the honeycomb becomes unstable and the bearing capacity is reduced. Finally, compaction occurs between the cells, which strengthens the material as the bearing load increases. Comparatively, the stress–strain curve of the GAHP is different [12]. The GAHP is prone to cracking in the double-layer area, reducing the bearing capacity of the structure. Because of the adhesive damage in the middle, the compression load capacity of the GAHP is far less than that of the BAHP.

### 3.2. Out-of-Plane Compression Test

In addition to out-of-plane compression properties, the in-plane compression properties of the BAHP also require further study. The test referred to standard GBT1454-2005, "Test method for edgewise compressive properties of sandwich constructions." The lengths, widths, and thicknesses of the samples were 120, 60 and 50 mm each, respectively, and the thicknesses of the single-cell wall and double-brazed-cell wall were 0.2 and 0.6 mm, respectively. The test was performed on an MTS tester at a loading rate of 0.5 mm/min. Owing to the differences in the properties in the longitudinal and transverse directions of the honeycomb, two honeycomb directions were chosen for the in-plane compression test. The load–displacement curves and the failure modes are presented in Figure 4.

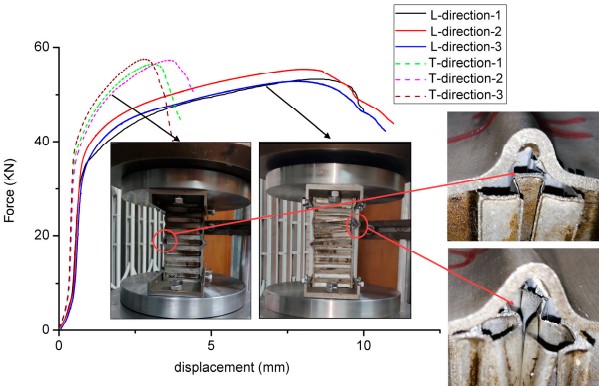

**Figure 4.** Load–displacement curves and failure modes of the BAHP under in-plane compression.

Overall, the transverse and longitudinal compressions are divided into three stages: elastic deformation, plastic deformation stage, and overall instability. The two elastic phases are close to each other, and the transverse compression has a larger modulus and a shorter stroke in the plastic deformation stage than longitudinal compression. The final failure mode shows that a honeycomb collapse and plate buckling occur with the longitudinal compression, which causes serious buckling of the plate in the cracking area only. However, under transverse compression, the honeycomb does not collapse remarkably, and the plate exhibits corrugated buckling. Because the plate is the main load-bearing part, the overall trends of the load–displacement curves are almost the same in both the directions. When longitudinally compressed, the single-cell wall primarily bears the load, causing core collapse with serious fractures. The plate is also severely deformed owing to the low strength. In comparison, under transverse compression, because the double-brazed-cell wall is thick, the welding connection strength with the panel is high, resulting in little collapse. Above all, the transverse compressive strength is slightly higher than the longitudinal compressive strength.

When GAHP is under in-plane compression [14], the honeycomb is weakly bonded to the panel. When the maximum load is reached, a large area debonds, and the load drops dramatically. By contrast, the BAHP has a relatively smoother load–displacement curve owing to the stronger interface strength.

## 4. Theoretical and Simulation Analysis

### 4.1. Method to Predict Out-of-Plane Compressive Strength

Considering the differences in the properties of a single-cell wall and a double-brazed-cell wall, a simplified strength theory for predicting the out-of-plane compressive strength of the BAHP is proposed. By comparing the tensile stress–strain curves of the single-cell and ten-brazed-cell wall presented in Section 2, it can be seen that the single-cell wall material of the hypotenuse wall and the brazed-cell wall material of the transverse wall demonstrate different constitutive relationships, yield strengths, and ultimate strengths. The critical factor of strength is the core performance; therefore, it is necessary to improve the traditional strength prediction method.

First, a single-cell wall is extracted as a four-sided simply supported plate. The compression failure of the plate can be considered based on three criteria: the stress reaching the critical instability stress, the maximum average stress after the instability, and the stress reaching the material compressive strength. According to reference [18], the instability critical stress ($\sigma_{cr}$) of a four-sided simply supported plate with one direction compressed is as follows:

$$\sigma_{cr} = k \cdot \frac{\pi^2 E}{12(1 - v^2)} \left(\frac{t}{b}\right)^2 \tag{1}$$

where $t$ and $b$ are the plate thickness and width of the compression edge, respectively; $E$ and $v$ are the compressive modulus and Poisson's ratio, respectively; and $k$ is the buckling coefficient. When the uncompressed edge is under a fixed constraint:

$$k = 0.83 - 0.93v + 1.34\left(\frac{\lambda}{\pi b}\right)^2 + 0.1\left(\frac{\pi b}{\lambda}\right)^2 \tag{2}$$

where $\lambda$ is the half-wavelength of buckling and $k = 1.3$, as calculated.

Because of the different properties of the single- and double-brazed-cell wall, their critical stresses $\sigma_{cr\ single}$ and $\sigma_{cr\ double}$ should be calculated separately. The limit load ($P_{pb}$) of the four-sided simply supported plate is expressed as:

$$P_{pb} = Ct^2 \sqrt{\sigma_B E} \tag{3}$$

where $\sigma_B$ is the compressive strength of the material. As the aluminum alloy is an isotropic material, the tensile strength obtained from the test is used as the compressive strength by default. $C$ is a parameter related to the material properties and sheet size. According to the literature test data [19], $C = 1.4–1.8$.

Therefore, the maximum average stress $\sigma_{pb}$ after the instability is expressed as:

$$\sigma_{pb} = C\sqrt{\sigma_B E}\left(\frac{t}{b}\right) \tag{4}$$

By observing the failure process of the honeycomb during compression, it can be inferred that the single-cell wall initially becomes unstable after reaching the critical stress, and subsequently, the excess stress that it cannot withstand is transferred to the transverse double-brazed-cell wall. This can be expressed as $\sigma_{cr\ double} > \sigma_{cr\ single}$, which is consistent with the observed phenomenon. In general, the single-cell wall will continue to bear the load until the stress reaches the maximum average stress. Following this, the load on the double-cell wall will continue to increase until the stress reaches its maximum average. Because the double-cell wall of the BAHP will have a decreased yield stress and modulus after brazing, it is necessary to consider whether the bearing capacity of the double-brazed-cell wall is weaker than that of the single-cell wall. Under the condition of good welding quality, the calculation yields $\sigma_{pb\ double} < \sigma_{pb\ single}$. Therefore, the maximum stress that the cell wall can reach is $\sigma_{pb\ single} = C\sqrt{\sigma_{B\ single} E_{single}}\left(\frac{t_{single}}{b}\right)$. Substituting the material parameters of the single-cell wall, we can obtain $\sigma_{pb\ single} = 140.3–180.4$ MPa.

Returning to the sandwich structure, according to the inverse proportion of the compression stress on the panel and core, the calculation formula for the out-of-plane compressive strength is:

$$\sigma_{zmax} = \frac{(d+c)}{(d+c\cos\theta)\sin\theta}\left(\frac{t}{c}\right)\sigma_{cmax} \tag{5}$$

where $\sigma_{cmax}$ is the maximum stress that the cell wall can withstand and $d$ and $c$ are the horizontal and hypotenuse lengths of the core, respectively, as displayed in Figure 5. The honeycomb is a regular hexagon, therefore, $c = d$ and $\theta = 60°$.

Calculating the critical stress and the maximum average stress of the single- and double-brazed-cell walls, respectively, yields the out-of-plane compressive strength of the structure as 7.18–9.23 MPa, which is consistent with the average strength of the test, 8.65 MPa. This strength prediction formula assumes there is no debonding in the double-brazed-cell walls of the honeycomb. Because of the excellent properties, the BAHP only buckles during compression and no debonding is observed.

When applying the formula, materials of the different wall thicknesses of the honeycomb are considered as different, and their critical stresss and maximum average stress are calculated, respectively; therefore, highly accurate results can be obtained. For in-plane compression, the existing strength prediction method [14,20] for GAHPs is not suitable for BAHPs, and hence, follow-up research is needed.

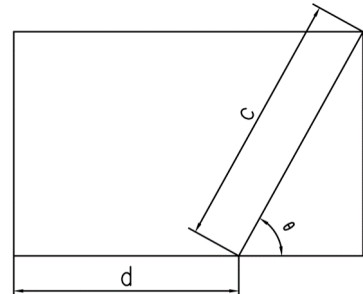

**Figure 5.** Planform of a honeycomb unit.

### 4.2. Finite Element Model

In the FEA, the elastoplastic material, MAT 24, is used to simulate BAHPs with single- and double-brazed-cell walls in LS-DYNA. This model constrains the lower rigid plate and applies a uniform-speed load to the upper rigid plate. The panels are constrained by U-shaped clamps above and below consistant with the test, and the clamps are in contact with the rigid plates. The contact between rigid plates, clampes the honeycomb panels is defined by *Contact_ Automatic_ surface_ to_ surface. The loading speed of the upper rigid plate is 500 mm/s, and the step size calculated is $1 \times 10^{-7}$/s.

Combined with experiments, the finite element model presented in Figure 6a,b. can be obtained. The in-plane models are divided into different directions of L-direction and T-direction. To verify the influence of the material properties on the calculated results, different models with single- and double-brazed-cell walls were built separately. One model applies single-cell wall material properties shown in Figure 2a, called as sing-cell wall model. Another applies ten-cell wall properties shown in Figure 2b, called as ten-cell wall model. The last one applies both the properties on the single-cell and double-brazed-cell wall respectively, called as mixed model. The material parameters of panel and core are shown in Table 2, the constitutive relation in the plastic stage is input to the card of MAT24 by stress-strain curve.

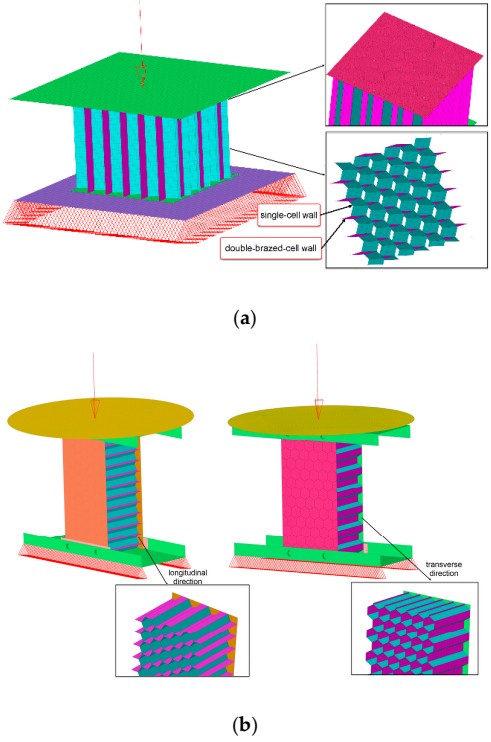

**Figure 6.** (**a**) Finite element model under out-of-plane compression; (**b**) Finite element model under in-plane compression in transverse and longitudinal directions.

**Table 2.** Material parameters of honeycomb.

| Material | Brand | Elastic Modulus | Poisson's Ratio | Yield Stress | Ultimate Stress | Density Kg/m³ |
|---|---|---|---|---|---|---|
| Panel | 6A02 | 72,327 MPa | 0.33 | 118 MPa | 180 MPa | $2.7 \times 10^3$ |
| Single-cell | 3003 | 58,710 MPa | 0.33 | 154 MPa | 160 MPa | $2.7 \times 10^3$ |
| Double-brazed-cell | 3003 | 105,250 MPa | 0.33 | 25 MPa | 134 MPa | $2.7 \times 10^3$ |
| Pressure head and fixture | \ | 206,000 MPa | 0.3 | 345 MPa | 600 MPa | $7.8 \times 10^3$ |

*4.3. Simulation Results*

The calculated results are displayed in Figures 7 and 8a,b.

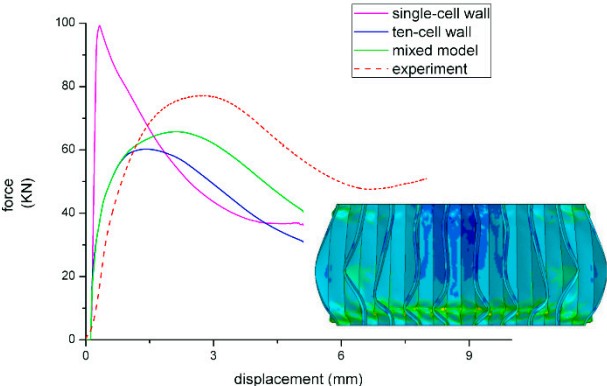

**Figure 7.** Simulation results with different models under out-of-plane compression.

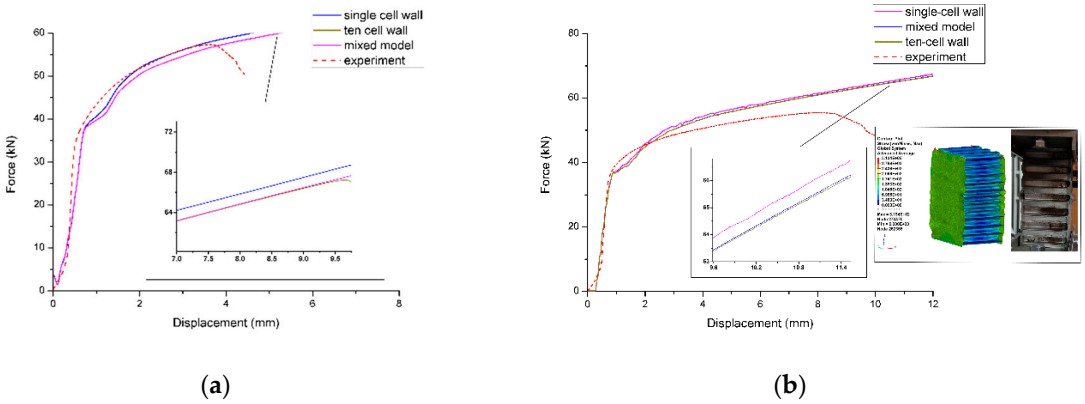

(**a**)                                     (**b**)

**Figure 8.** (**a**) Results of the BAHP under in-plane compression in the transverse direction; (**b**) Results of the BAHP under in-plane compression in the longitudinal direction.

As Figure 7 shows, the elastic modulus of the single-cell wall model is extremely large, and the peak load of the double-cell wall cannot reach the test level. Simultaneously, the model considering two material properties can effectively simulate the compression properties of the BAHP. The deformation results of the different models are similar. Because of the significant influence of the cell wall thickness on the compression properties, because of errors in the measurement, the simulation and test results cannot be completely fitted.

The simulation results under in-plane compression in the transverse and longitudinal directions are presented in Figure 8a,b. It is observed that the curves from the elastic deformation stage to the plastic deformation stage can be fitted well, whereas the third stage, i.e., the overall instability stage, has significant discreteness, because its turning point depends on the defect location. As the finite element model does not consider the defects of manufacturing, the maximum load value is slightly higher than that in the test. Irrespective of the direction, compared to the calculated results, the load

is slightly higher for the single-cell wall model. The load–displacement curves are generally very close. When the instability approaches the back segment, the calculated results show little differences. It can be inferred that during the in-plane compression, the characteristics of the early stage are mainly determined by the panel, with the core material playing only an auxiliary role. Until the core is close to collapse and the entire structure becomes unstable, the honeycomb core affects the calculated results.

From the perspective of deformation, as Figure 8b demonstrates, the panel will be wrinkled under in-plane compression, which is basically consistent with the test results. Without considering the manufacturing defects and the cracking of the panel and core, the deformation does not fully fit the test results. The impact of defects on FE simulation results will be analyzed in detail in the following sections.

According to the shell theory, the equivalent modulus of laminate is obtained by [21]:

$$E_{eq} = \frac{1}{\frac{V_c}{E_c} + \frac{V_f}{E_f}} \tag{6}$$

where $E_f$ is the panel modulus, $E_c$ is the core modulus, $V_f$ and $V_c$ are the volume fraction of panels and core. As the value of $E_f \gg V_f$, so the value of $V_f/E_f$ is very small. $E_{eq}$ depends on $E_c$.

Assuming that the interface between the panel and the honeycomb core is well connected, the equivalent bending stiffness of the sandwich structure is [14]:

$$(EI)_{eq} = E_c \cdot \frac{bt_c^3}{12} + E_f b \cdot \left( t_c t_f^2 + \frac{t_c^2}{2} t_f + \frac{2}{3} t_f^3 \right) \tag{7}$$

where $t_f$ is the panel thickness, $t_c$ is the core thickness, $b$ is the width of the structure. As $E_c \cdot \frac{bt_c^3}{12}$ is the core stiffness contribution to $(EI)_{eq}$, $E_f b \left( t_c t_f^2 + \frac{t_c^2}{2} t_f + \frac{2}{3} t_f^3 \right)$ is the panel stiffness contribution to $(EI)_{eq}$, which is relatively large, combining with simulation results, we can draw the conclusion that the out-of-plane compression properties depend on the core strongly, which has a major impact on the calculated results with changing core material parameters. However, under in-plane compression, the load–displacement curve depends on the panel, and the core parameter change has little influence on the results.

### 4.4. Defects Analysis

During the manufacture of honeycomb plates, defects are inevitable and there are many studies on defect models [8,22,23], including honeycomb shape defects, missing struts for the unit cell, debonding between panels and core, etc. Based on the specimen characteristics, the defect analyses are focused on the uneven height of the panels and the debonding between core and panels shown in Figure 9. As Table 3 shows, there are four defect types and defect levels, and each defect type defines three defect levels. The FE model is the T-direction in-plane compressive model, except for the defect part, the boundary conditions and material parameters are unchanged.

The defects of uneven height are divided into unilateral with dh = 0 mm, 1.5 mm and 2.5 mm and bilateral with dh1 = 1.5, dh2 = 0 mm, 0.5 mm and 1 mm. The debonding defects between panels and the core are also divided into unilateral and bilateral, and the level is one row, two rows and three rows of unit cell debonding.

As Figure 10a,b shown, the uneven height lengthens the contact time between rigid plate and the honeycomb plate, which has little influence on the elastic stage and greater influence on the plastic plastic stage, and shortens the overall instability time. Figure 10c,d are the simulation results of unilateral and bilateral debonding. The ideal model is steadily compressed with a long time to collapse. However, debonding will make the model unstable in advance. The greater debonding level, the earlier instability will occur. All defects will lead to a reduction in structural strength and impact the final failure mode.

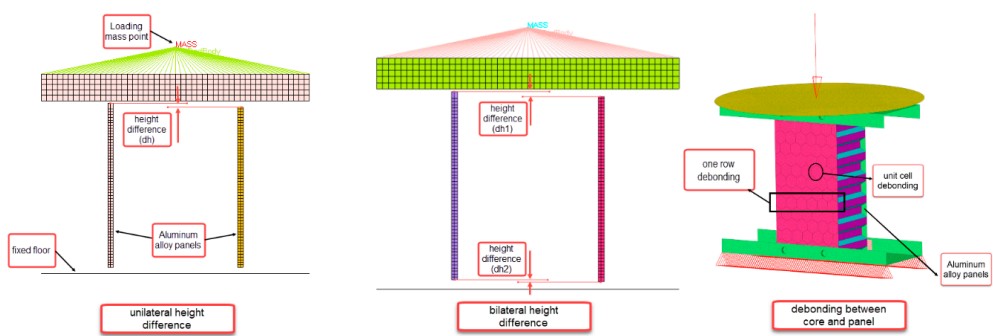

**Figure 9.** Schematic diagram of defects in aluminum honeycomb panel.

**Table 3.** Defect types and defect level.

| Defect Types | Defect Levels |
|---|---|
| Panel unilateral uneven height | dh = 0 mm |
| | dh = 1.5 mm |
| | dh = 2.5 mm |
| Panel bilateral uneven height | dh1 = 1.5 mm, dh2 = 0 mm |
| | dh1 = 1.5 mm, dh2 = 0.5 mm |
| | dh1 = 1.5 mm, dh2 = 1 mm |
| Unilateral debonding between panels and core | One row of unit cell debonding |
| | Two row of unit cell debonding |
| | Three row of unit cell debonding |
| Bilateral debonding between panels and core | One row of unit cell debonding |
| | Two row of unit cell debonding |
| | Three row of unit cell debonding |

(**a**)

(**b**)

(**c**)

(**d**)

**Figure 10.** (**a**) Force-time curve with panel unilateral uneven height; (**b**) Force-time curve with panel bilateral uneven height; (**c**) Force-time curve with unilateral debonding; (**d**) Force-time curve with bilateral debonding.

## 5. Conclusions

In this study, by conducting out-of-plane and in-plane compression tests on a BAHP, comparing its properties with those of a traditional GAHP, and combining theoretical and finite element analysis, the following conclusions can be drawn:

(1) Tensile tests of the single-cell wall and ten-brazed-cell walls in the honeycomb were conducted, showing that their material properties are completely different. Therefore, their differences should be mainly considered when studying a BAHP.

(2) Considering the out-of-plane compression properties of the BAHP, the brazed area is extremely tight such that there is no debonding. Regarding the in-plane compression properties, two compression directions were designed: transverse and longitudinal. The load–displacement curves, ultimate loads, and failure modes are different in different directions. The interface bonding strength of the BAHP is high and there is no large-scale debonding between the core and the panel.

(3) A theoretical method for the prediction of the strength of the BAHP under out-of-plane compression is proposed. Considering the properties of the BAHP, this method can yield accurate results. The results of the simulation considering two cell walls are more consistent to the test. From the simulation and theoretical analysis, it was seen that the core material parameters have a significant influence during out-of-plane compression, whereas the in-plane compression properties mainly depend on the panel. With the uneven height and debonding defects, the FE models are earlier entered the stage overall instability which are more close to the test.

There are few studies on BAHPs, and their bending, shear properties, and impact response as well as the effect of defects on mechanical properties still need to be further studied.

**Author Contributions:** Investigation, L.J., G.Y., T.Z. and D.D.; project administration, B.Y. and S.X.; simulation analysis, J.Z. and R.L.; supervision, B.Y.; writing—original draft, L.J.; writing—review & editing, B.Y. and S.X. All authors have read and agreed to the published version of the manuscript.

**Funding:** This research was funded by National Key R&D Program of China (2016YFB1200602-14); the National Natural Science Foundation of China (51675446); and the Independent Subject of State Key Laboratory of Traction Power (2020TPL-T07).

**Acknowledgments:** The authors would like to thank Juan Zhang at Southwest Jiaotong University for her help during the experiment.

**Conflicts of Interest:** The authors declare no conflict of interest.

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
