# Peer review of "Experimental and Simulation Studies on the Compressive Properties of Brazed Aluminum Honeycomb Plates and a Strength Prediction Method"

_metals, doi:10.3390/met10111544_

Round 1
Reviewer 1 Report
The paper deals with experimental tests of Al honeycomb plate and numerical simulations to describe the experimental compression tests.
The paper contains interesting results, but the discussion of numerical models it is very weak. The manuscript can be considered, if authors are ready prepare a new version with a major revision of the manuscript.

Author Response
Response to Reviewer 1 Comments
Point 1:The paper is badly organized and a section should be dedicated to description and assumptions of numerical models and another one to numerical results;
Response 1:Assumptions of numerical models and the numerical results are divided to section 4.2 and section 4.3. More details of the model and simulation results have also been added in corresponding section.
Point 2:you can try to model the buckling resistance under in-plane compression re-ferring the beams and shell theory. In details, your conclusion that "under in-plane compression, the load– displacement curve depends on the panel, and the core parameter change has little influence on the results." can be antici-pated by simple analyses;
Response 2:At the end of section 4.3, relevant analysis of shell theory and beam buckling theory is added. It is verified by theoretical analysis that the out-of-plane compression properties depend on the core strongly, the out-of-plane compression properties depend on the panel.
Point 2: It is not clear the difference among the different models that the authors are using:
(1)Fig. 7 only represent in-plane models but the following figure refer to results for out-of-plane compression;
(2)in the text you do not define what is the mixed model that appears for the first time in the legend of Fig. 8 for out-of-plane compression;
(3)referring to in-plane compression there is no description of the models in the text and it is not understood what is the ten- cell wall;
Response 3:
- out-of-plane compressive models are shown in Fig. 6(a), and more details have also been added to the image to point out the sing-cell wall and double-brazed cell wall.
(2)the mixed model and two other models are presented in section 4.2 of line 245-248. The mixed model input the material properties respectively with single-cell wall stress-strain curve obtained as Fig 2(a), and double-brazed-cell wall stress-strain curve obtained as Fig 2(b).
(3)More description of the models are added in section 4.2 of line 245-248. To study the changes in the core material after brazing, we designed the ten-brazed-cell wall specimen, and obtained the stress-strain curve under tensile load in Fig.2. More information is also added in the first paragraph of section 2.2.
Point 4: the mention to manufacturing defects should be supported by literature and some sensitivity study by FE simulations.1
Response 4: The literature of [8], [22] and [23] are added to support the defect analysis. A new part of section 4.4 is added in the manuscript. In this section, four different defect types and three defect levels are analyzed by FE simulations. The a large number simulation results and conclusions are also described in 4.4.
Or Please see the attachment.

Reviewer 2 Report
Review
This work can be published in the journal, but there are a number of comments:
- The work does not show the composition of the aluminum alloy. It is necessary to write the composition of the alloy.
- The work uses soldering. The composition of the solder, its type (powder, tape, etc.) and the soldering mode (temperature, soldering time, pressure, external conditions) are not shown. It is necessary to write this data
- Figure 2 must be divided into a) and b) and signed
- Figure 4 shows the L-direction and T-direction. It is necessary to decipher the notation in the text
- The work lacks metallographic analysis of the destruction zones of the samples. This would strengthen the work.
Author Response
Response to Reviewer 2 Comments
Point 1:The work does not show the composition of the aluminum alloy. It is necessary to write the composition of the alloy.
Response 1: The composition of the aluminum alloy is shown in Table 1 respectively by panel and core in section 3.1.
Table 1. The material compositions of panels and core
|
construction |
brand |
Si |
Cu |
Mn |
Zn |
Fe |
Al |
|
panel |
A602 |
0.5%~1.2% |
0.2%~0.6% |
0.15%~0.35% |
0.2% |
0.5% |
the rest |
|
core |
3003 |
0.6% |
0.05%~0.2% |
1.0%~1.5% |
0.1% |
0.7% |
the rest |
Point 2:1.The work uses soldering. The composition of the solder, its type (powder, tape, etc.) and the soldering mode (temperature, soldering time, pressure, external conditions) are not shown. It is necessary to write this data
Response 2: The soldering process is conducted by another laboratory which would not give further information, our team is focus on the mechanic properties. More details like soldering time and pressure are the confidential technology. As far as we known, the solder is kind of powder, the temperature is 600℃, and there is a special brazing furnace. Known information has been added to the first paragraph of section 2.1.
Point 3:Figure 2 must be divided into a) and b) and signed.
Response 3: Figure 2 has been divided into a) and b) and signed in the manuscript.
Point 4:1.Figure 4 shows the L-direction and T-direction. It is necessary to decipher the notation in the text.
Response 4: L-direction and T-direction are represent longitudinal direction and transverse direction. The explanation is added in the first paragraph of section 2.1. The schematic diagram of the direction in Figure 1 is updated too.
Point 5:The work lacks metallographic analysis of the destruction zones of the samples. This would strengthen the work.
Response 5: In the out-of-plane compressive test, the failure mode showed as the folding of the honeycomb without obvious fracture. In the in-of-plane compressive test, the failure mode showed as the debonding between panels and core. The debonding area is difficult to do the metallographic analysis. To better display the destruction zones, Figure 3 and Figure 4 added more details in the manuscript.
Or Please see the attachment.

Round 2
Reviewer 1 Report
The paper has now improved. Perhaps a better description of what is the 'mixed model' would be helpful (you simply mix the simulated response of single cell and 10 cells) ?
Author Response
piont 1: The paper has now improved. Perhaps a better description of what is the 'mixed model' would be helpful (you simply mix the simulated response of single cell and 10 cells) ?
response 1: In previous studies, when the researchers simulate the honeycomb core, they do not consider the material property differences between single-cell wall and double-cell wall. After brazing, the material properties of double-cell wall have been greatly changed. In the mixed model, we assigined single-cell wall and double-cell wall with different parameters. The stress-strain curve of double-cell wall was obtained by the tensile test in section 2.2.